# Anti-Melanogenic Potential of Natural and Synthetic Substances: Application in Zebrafish Model

**DOI:** 10.3390/molecules28031053

**Published:** 2023-01-20

**Authors:** Adriana M. Ferreira, Agerdânio A. de Souza, Rosemary de Carvalho R. Koga, Iracirema da S. Sena, Mateus de Jesus S. Matos, Rosana Tomazi, Irlon M. Ferreira, José Carlos T. Carvalho

**Affiliations:** 1Research Laboratory of Drugs, Department of Biological and Health Sciences, Federal University of Amapá, Rod. JK, km 02, Macapá 68902-280, Amapá, Brazil; 2Laboratory of Biocatalysis and Applied Organic Synthesis, Department of Exact Sciences, Chemistry Course, Federal University of Amapá, Rod. JK, km 02, Macapá 68902-280, Amapá, Brazil; 3Federal Institute of Amapá, Chemistry Course, BR-210 Highway, km 03, S/N—Brasil Novo, Macapá 68909-398, Amapá, Brazil

**Keywords:** melanogenesis inhibitors, tyrosinase, melanin, danio rerio

## Abstract

Melanogenesis is a biosynthetic pathway for the formation of the pigment melanin in human skin. A key enzyme in the process of pigmentation through melanin is tyrosinase, which catalyzes the first and only limiting step in melanogenesis. Since the discovery of its methanogenic properties, tyrosinase has been the focus of research related to the anti-melanogenesis. In addition to developing more effective and commercially safe inhibitors, more studies are required to better understand the mechanisms involved in the skin depigmentation process. However, in vivo assays are necessary to develop and validate new drugs or molecules for this purpose, and to accomplish this, zebrafish has been identified as a model organism for in vivo application. In addition, such model would allow tracking and studying the depigmenting activity of many bioactive compounds, important to genetics, medicinal chemistry and even the cosmetic industry. Studies have shown the similarity between human and zebrafish genomes, encouraging their use as a model to understand the mechanism of action of a tested compound. Interestingly, zebrafish skin shares many similarities with human skin, suggesting that this model organism is suitable for studying melanogenesis inhibitors. Accordingly, several bioactive compounds reported herein for this model are compared in terms of their molecular structure and possible mode of action in zebrafish embryos. In particular, this article described the main metabolites of *Trichoderma* fungi, in addition to substances from natural and synthetic sources.

## 1. Introduction

Estimates indicate that approximately 15% of the world’s populations invest in skin whitening [1] with melanogenesis as one of the main reasons. Melanogenesis is a complex process with different physiological stages. Any imbalance in this process can cause different types of pigmentation deficiency, classified as hypopigmentation or hyperpigmentation, and may occur with or without changes in the number of melanocytes. The serious pathological consequence of such physiological imbalance is cancer derived from melanocytes, or melanomas, which are among the most aggressive, metastatic, and lethal forms of skin cancer [2].

In addition to physiological imbalance, other factors can inhibit melanin production. These include pharmaceutical or cosmetic additives, which cause adverse side effects, such as skin irritation, cytotoxicity, and carcinogenicity. In addition, because of the low stability of some formulations and low penetration into the skin, their multiple use must be limited [3]. Moreover, studies reported that many have been linked to neurodegenerative diseases, including Parkinson’s, Alzheimer’s, and Huntington’s diseases [4,5,6,7].

Melanogenesis occurs in melanocytes, which are found in the basal layer of the epidermis, with tyrosinase as the unifying biochemical characteristic of melanogenesis in plants and animals. Tyrosinase (EC 1.14.18.1), a copper oxidase, is a type 3 copper containing metalloenzyme widely distributed in bacteria, fungi, insects, plants, and animals, including humans, to produce melanin pigments [8]. Initially, tyrosinase is synthesized on the surface of the rough endoplasmic reticulum. It is considered a key and limiting enzyme for the in vivo synthesis of melanin. Melanin plays a key role in several biological functions, including the pigmentation process of mammalian dermis. As a component of primary immune response, it is a triggering agent of the wound healing system in plants and fungi [9].

The effect of tyrosinase is observed in various living organisms. In plants, it is observed in degradation processes. In fungi, tyrosinase acts in the differentiation of reproductive organs during spore formation [10,11,12]. In humans, tyrosinase, which regulates melanin, is responsible for the coloration of the skin, eyes, and hair, with high diversity among human populations [13,14,15].

The discovery of new molecules from natural products and fungal extracts which have anti-melanogenesis activity is ongoing. Since these molecules would be expected to minimize the side effects of pigmentation treatment, they represent a potent, low-cost and effective alternative [16,17,18,19].

Fungal metabolites have stood out as substantial melanogenesis inhibitors owing to their pharmacological potential [20]. Thus, fungi of different genera, which demonstrate anti-melanogenic activity with antibiotic action, and growth regulators in vegetables and fruits, among others, have attracted the interest of researchers who are pursuing the discovery and isolation of new compounds in the agricultural, food and pharmaceutical industries [21]. 

However, according to the World Health Organization (WHO), researchers must follow the NEQ (Needs Evaluation Questionnaire) validation process when carrying out pharmacological tests in traditional in vivo and in vitro systems in order to increase investment in research and innovation, mainly in underdeveloped countries.

Several in vivo models have been used extensively to investigate anti-tyrosinase mechanisms [22]; however, some are limited from a practical point of view and others from a physio/pathological point of view. Consequently, researchers have resorted to emerging models, such as zebrafish (*Danio rerio*). This model has the advantages of small size, ease of handling and maintenance, and rapid reproduction rate, as well as the high efficiency of drug penetration through skin and gills [23,24]. Moreover, zebrafish have a fully characterized genome with functional domains of many key proteins nearly identical to their human homologues [25,26].

In addition, the use of the zebrafish model has enabled the development of new approaches, the refinement of techniques, and the insertion of quantitative and qualitative parameters into the screening of bioactive compounds based on phenotypes. In particular, zebrafish analysis has been linked to the presence or absence of melanin since the pigmentation process can be observed on the surface of the zebrafish embryo without complicated experimental procedures [27,28,29,30]. 

Therefore, this work aimed to review the most recent scientific information available on melanogenesis inhibitors of natural (plant or fungal) and synthetic origin using zebrafish as an experimental model. An important part of the review involves clarifying how the zebrafish depigmenting system works and whether it resembles that of humans. The inclusion criteria for this study were original articles exclusive to the genus and species studied with full text available in portuguese, english or other languages. Exclusion criteria included abstracts, online sites without scientific sources, incomplete text, and unrelated or repeated articles, according to the methodology previously described [31].

The descriptive words used in our search were (a) *Trichoderma* spp. and their secondary metabolites; correlated to the potential, (b) anti-melanogenic agent, (c) Tyrosine, (d) natural and synthetic products in the zebrafish model.

## 2. Melanin and Tyrosinase Mechanism of Action

Melanin is synthesized by melanocytes, which are directly related to neighboring keratinocytes. It is an amorphous substance formed by the polymerization of phenolic and indole compounds. Specific to the skin, melanin protects the epidermis against harmful stimuli, such as UV-radiation, through melanogenesis, the process regulating autocrine or paracrine factors, including α-melanocyte-stimulating hormone and endothelin. Together with this intricate system, keratinocytes and skin cells, such as fibroblasts and immune cells, are regulators of the behavior of melanocytes, which, in turn, are produced by paracrine factors.

This series of reactions makes the polymerized material available spontaneously as melanin. The formation of melanin is dependent on the catalysis of L-tyrosine in L-DOPA, but not intermediate dopachrome, also called (TRP2), though both are direct products of the tyrosinase cycle (Figure 1).

Melanin is an amorphous polymer, negatively charged, but derived from the auto-oxidative polycondensation of several quinone groups with hydrophobic properties [32]. Therefore, the pathway of melanogenesis (Figure 1) can be conveniently divided into two phases: proximal, which consists of the enzymatic oxidation of a monophenol (tyrosine) and/or o-diphenol (L-DOPA), to its corresponding *O*-quinone and distal, which is represented by chemical and enzymatic reactions occurring after the formation of dopachrome to direct the synthesis of eumelanins, which are either derived from DHICA (5,6-dihydroxyindole-2-carboxylic acid; brown) or from DHI (5,6-dihydroxyindole; black) [33,34,35,36,37]. 

Therefore, synthesis of eumelanin is directly linked to the process of melanin pigments responsible for retaining the ability to deactivate free radicals, peroxides and absorb heavy metals and toxic electrophilic metabolites, thus exhibiting strong antioxidant activity in addition to absorbing light in a wide spectrum range including UV [3,38]. By deregulating this system, hyperpigmentation can occur. This is equivalent to tyrosinase hyperactivity, which is normally associated with pathological disorders, such as spots, melasma and the appearance of melanomas [39]. Therefore, it is fundamentally important to regulate tyrosinase productions so that balance in the melanogenesis process is maintained and pathogenicity is avoided [40,41].

## 3. Inhibitors of Melanogenesis by Fungi of the Genus Trichoderma 

The literature presents several potential tyrosinase inhibitors, both from natural and synthetic sources. However, studies that investigate the molecular and functional characterization of this enzyme are rare, mainly those specific to anti-melanogenic activity originating from organisms, such as heterotrophs. In this sense, fungi stand out for their potential, with greater occurrence in the genus *Trichoderma* (Hypocreales, Ascomycota), having more than 300 species with high adaptive capacity, favoring their presence in different natural environments under different climatic conditions [42,43,44].

Species of the genus *Trichoderma*, order Hypocreales, have greater phytogeographic occurrences in the soil of regions with a humid tropical climate. Such conditions produce a class of Hyphomycetes characterized as filamentous and cosmopolitan fungi with diverse biotechnological applications [43,45,46]. Given these characteristics, various species of *Trichoderma* use a wide variety of compounds as a source of carbon and nitrogen, a typical characteristic of fungi from saprophytic soil [47]. 

Drawing on their chemodiversity, tyrosinase inhibitors biosynthesized by fungi are derived from isoflavones and pyrones, along with terpenes, steroids, and alkaloids, which can reversibly or irreversibly inactivate the enzyme [48]. 

In particular, some studies report *Trichoderma reesei* and *Trichoderma harzianum* as significant producers of extracellular tyrosinase, previously characterized, isolated and purified by precipitation with ammonium sulfate (85%). Purified tyrosinase exhibited a final specific activity of 69.39 and 65.11 U/mg of protein, values which double the purification of 21.09 and 14.93 for *T. reesei* and *T. harzianum,* respectively [49,50].

Other studies described the activity of fungal extracts of *Trichoderma atroviride*, *Trichoderma gamsii*, *Trichoderma guizhouense* and *Trichoderma songyi*. These extracts were reported to have tyrosinase inhibitory capacity associated with the elimination of reactive quinone products. Furthermore, a trichoviridine cyclopentyl isocyanide, MR566A and MR566B, isolated from *T. reesei*, showed moderate cytotoxicity against the human melanoma cell line A375-S2 [51].

Studies show that antioxidant activity is associated with the ability to inhibit tyrosinase. However, fungal extracts of *Trichoderma atroviride*, *Trichoderma gamsii*, *Trichoderma guizhouense* and *Trichoderma songyi*, of marine origin, which showed a considerable ability to inhibit tyrosinase (IC 50 < 100 μg/mL), demonstrated low radical scavenging activity (<50%). This suggests the presence of other mechanisms that inhibit tyrosinase, such as competitive inhibitors, including copper chelators that inhibit this metal coenzyme, or suicide inhibitors that inactivate tyrosinase by altering tertiary and quaternary structures of the enzyme [52]. 

Viridiofungins, broad spectrum antifungal agents, are derived from the secondary metabolite of *Trichoderma viride*. They act as inhibitors of tyrosinase and farnesyl transferase and the farnesylation of the oncogenic Ras protein, indicating their potential to treat cancer [53]. Furthermore, an oxazole derivative called melanoxazal, which is isolated from the fermentation broth of *Trichoderma strain* ATF-451, showed strong inhibitory activity against mushroom tyrosinase [54].

A strain of *T. harzianum*, an isomer designated as MR304A, was isolated and identified as an isocyanide compound, demonstrating inhibition of melanin formation in *Streptomyces bikiniensis* and B16A melanoma cells [55]. Still related to *T. harzianum* isolated from soils, the authors verified the inhibition of melanin synthesis by two new tyrosine inhibitors. MR566A, along with a new oxazole compound, MR93B, exhibited activity similar to that of MR93A. in addition to isocyanide compounds, identified as derivatives of alkyl citrate (Table 1) [56,57], a group of isocyanide compounds acting in the inhibition of tyrosinase activity [20]. 

The strain *Trichoderma viride* H1-7 from a marine environment presented a tyrosinase inhibitory factor through the Homothallin II structure (Table 1). It was preliminarily isolated and studied as an antibiotic from *Trichoderma koningii* and *T. harzianum* [58]. These fungi are excellent producers of extracellular enzymes, and seven new molecules were isolated from the metabolites of these species, demonstrating anti-melanogenic activity by binding to the copper active site [48]. 

In addition to experimental research involving endophytic fungi with enzymatic activity against tyrosinase, the supernatant of the metabolite of *Trichoderma atroviride* has found an industrial use in the manufacture of functional whitening cosmetics, but it is also a potential inhibitor of tyrosinase [59]. 

## 4. Anti-Melanogenic Activity in Zebrafish Embryo

Assays involving the mechanisms of action of tyrosinase have become increasingly important for two reasons: (a) the elucidation of inhibitory pathways in melanin pigment synthesis; and (b) the growing demand for anti-melanogenic agents capable of reducing or inhibiting the unwanted side effects of current treatments [60]. Consequently, R&D efforts have turned to experimental models, such as zebrafish, able to facilitate the in vivo screening of anti-melanogenic agents. Such models are low in cost, but high in fertilization rate and genetic homogeneity, relative to mammalian models, thus enabling screening for tyrosinase-reactive drugs and cosmetics [60]. 

Compared to the zebrafish model, traditional models have both physiological and economic disadvantages. Therefore, experimental robustness and safety in zebrafish, as a phenotype-based screening model for melanogenic inhibitors or stimulators, have advanced considerably in recent years [61,62].

Because it is a model with biological similarity to more complex organisms, its genome shares more than 70% genes with humans [63,64]. This small teleost has three types of pigment cells: iridophores (containing reflective lines, blue), xanthophores (yellow) and melanophores, factors that have favored its use (Figure 2). The existence of homogeneity in the genetic characteristics of genes related to melanogenesis in zebrafish, such as TYR that gives instructions for making tyrosinase, which resemble mammalian genes, is decisive for the selection of this experimental model in anti-melanogenic studies [65,66].

The zebrafish model can be used to understand phylogeny mechanisms and TYR patterns expressed in melanophores relative to time. Specifically, the formation of pigmentation in zebrafish begins directly in the epithelium and pigment of the developing retina with subsequent transcription of TYR within 16.5 h post fertilization (hpf). Melanin in melanophores can be detected in the dorsolateral skin and retina at approximately 24 hpf. Because melanin is synthesized in melanophores in the early stages of zebrafish embryonic development, microscope-assisted observation is possible [30,61,66].

On the other hand, tyrosinase inhibitors derived from secondary metabolites of bacteria and fungi are known to produce anti-melanogenic compounds. Currently, many of these molecules have been identified (Figure 3) and tested for their anti-melanogenic activity in zebrafish [68,69].

Among the most commercially used tyrosinase inhibitors, kojic acid and its derivatives are derived from secondary metabolites produced by fungi of the genera *Aspergillus* and *Penicillium* [35,72]. These metabolites are used in the cosmetics industry for skin whitening, as well as a food additive to prevent enzymatic browning in the food oxidation process. However, they are also used as a standard in research involving melanogenesis inhibitory activity in the zebrafish model [3,37]. Kojic acid is hydrophilic and acts as a Cu^2+^ chelating agent at the active site of tyrosinase to suppress the tautomerization of dopachrome to 5,6-dihydroxyindole-2-carboxylic acid [68,73]. 

Ethanolic extract of *Laetiporus sulphureus* (LSE) and *Agaricus silvaticus* (ASE), edible mushrooms, underwent biochemical mapping for their anti-melanogenic effect and were found to effectively inhibit melanogenesis in a dose-dependent manner (400–500 µg/mL). However, the exploited extracts at the depigmenting dose did not show adverse effects on the melanocytes of zebrafish embryos [74]. 

To validate the in vivo anti-melanogenesis activity of *Antrodia cinnamomea* ethanol extracts, a study was based on the zebrafish phenotype. In experiments, the AC_Et50_Hex extract fraction exhibited depigmenting activity similar to that of kojic acid (56.1% vs. 52.3%), but with lower dosage (50 ppm vs. 1400 ppm), in addition to demonstrating less toxicity to embryos [75]. 

Studies which evaluated the ability of modified Shiitake extract (A37) and wild Shiitake extract (WE) demonstrated that A37 conferred less pigmentation in zebrafish embryos and inhibited the growth of melanoma cells better than WE. The difference in cell cycle profile suggests that the greater anticancer effect of the A37 extract results from changes in the metabolite produced as a result of mutation such that A37 is also capable of inhibiting GSK3β phosphorylation. Both extracts contain 14 compounds in common [76]. 

Azelaic acid [(1, 7-heptanedicarboxylic acid) Table 1], produced by *Pityrosporum ovale*, a strain found naturally in wheat, rye and barley [77,78], is often used for the treatment of acne, rosacea, skin pigmentation and freckles. The compound can bind to amino and carboxyl groups and prevent the interaction of tyrosine in the active site of tyrosinase, acting as a competitive inhibitor [73,79,80]. 

Interestingly, azelaic acid has demonstrated thioredoxin reductase inhibition in cultured human keratinocytes, melanocytes, melanoma cells, murine melanoma cells and purified enzymes from *Escherichia coli*, rat liver and human melanoma [77,81]. This may explain the antiproliferative and cytotoxic effect, the synthesis of deoxyribonucleotides. Moreover, azelaic acid, when combined with taurine, an antioxidant compound, inhibits tyrosinase by activating the ERK pathway [82,83].

## 5. Natural Products Used as Melanogenesis Inhibitors in Zebrafish

Melanogenic inhibitors 1-phenyl-2-thiourea, arbutin, kojic acid, 2-mercaptobenzothiazole and synthesized compounds (haginin, YT16i) [61] were used in zebrafish embryos, and the inhibitory effects on pigmentation were indicated. However, compound YT16i showed major abnormalities in terms of morphological deformities and cardiac function, along with high toxicity at higher concentrations (Table 2) [66].

Triclocarban (3,4,4′-trichlorocarbanilide) has TYR inhibition activity and is present in soaps, shampoos, cosmetic detergents, and toothpastes [62]. At a concentration of 50 µg/L, zebrafish embryos exposed to triclocarban showed signs of toxicity, such as mortality and a significant index of teratogenicity [62]. Based on this characteristic, several studies have already shown that the exposure of developing embryos to chemicals considered pollutants will cause the dysregulation of thyroid hormones, resulting in craniofacial and ocular pathologies [87,88]. 

Omeprazole reduces pigment area density in zebrafish embryos by 63% at a concentration of 60 µM. Furthermore, intracellular TYR activity was decreased by 48%, compared to untreated zebrafish embryo, after treatment with omeprazole [89]. 

Several studies on plant and fungal extracts used zebrafish as an in vivo experimental model to investigate tyrosinase inhibition or their activity as depigmenting agents. Among these studies, extracts from *Ecklonia cava* and *Sargassum siliquastrum* seaweeds showed slight toxicity. *Phlorofucofuroeckol-A* (PFF-A) isolated from a seaweed species, *Ecklonia cava*, demonstrated an attenuating effect against tyrosinase in the B16F10 cells of zebrafish embryos. When evaluating the safety and efficacy of PFF-A for anti-melanogenic effects, the study tested low doses of PFF-A (1.5–15 nM) [90]. This suggests that low doses of *E. cava* derived PFF-A can suppress embryonic pigmentation and melanogenesis. This indicates the possibility of using PFF-A as an anti-melanogenic agent [90]. 

Studies with extracts of the marine *Pseudomonas anoectochilus* and *P. narcissus* showed a potentiated effect in inhibiting zebrafish embryo TYR [85,91], and the natural compound derived from oleic acid, produced in the small intestine as oleoylethanolamide, reduced TYR by about 49.5% at a concentration of 150 µM in zebrafish embryos [92]. In contrast, sesamol, a bioactive lignan from *Sesamum indicum*, inhibited melanin biosynthesis in a concentration-dependent manner in zebrafish embryo. The absence of pigmentation can be explained by reduced TYR activity and gene expression related to melanogenesis [93] (Table 3).

## 6. Synthetic Compounds Used as Melanogenesis Inhibitors in Zebrafish

Low molecular weight synthetic compounds between 100 and 300 g/mol can, when subjected to anti-melanogenic activity in zebrafish, be divided into several categories and molecular size [113]. For example, phenylthiorea [86,96,114], sodium erythorbate [34], 2-methylphenyl-*E*-(3-hydroxy-5-methoxy)-styryl ether [98], 4-phenyl hydroxycoumarins [115], kojic acid palmitate [116] and MEK-I (Table 4) inhibited melanophores in zebrafish embryos. Since these compounds vary by molecular size, stability, hydrophilicity and hydrophobicity, they can permeate the membrane and accommodate bioavailability in the embryo depigmentation process [98]. 

Hydrophobicity is an important feature that demonstrates an affinity for permeating cell membranes. For zebrafish, several membrane layers must be taken into account, including chorion, melanocyte cell membrane and melanosome plasma membrane [70,121]. 

Chorion is a porous channel measuring 0.5 to 0.7 µm in diameter with a gap at intervals of 1.5 to 2.5 µm. It surrounds the embryo, thus reducing the rate of diffusion of small molecules in the embryo (Figure 3) [103,106]. Most of these compounds show conformity in the benzene ring structure with a varied number of hydroxyl groups (OH) bonded to it. This chemical feature may explain the cellular permeability that leads to TYR inhibition [70,122,123].

Twenty-seven new cinamides, consisting of cinnamic acid derivatives similar to 1-aryl piperazines, were synthesized and evaluated for potential tyrosinase inhibitory activity. Among them, 3-chloro-4-fluorophenyl moiety at the N-1 of the piperazine ring was essential for potent tyrosinase inhibitory effect with 3-nitrocinnamoyl and 2-chloro-3-methoxycinnamoyl. In general, all compounds characterized by the presence of 1-(3-chloro-4-fluorophenyl)piperazine () demonstrated the ability to inhibit melanogenesis in A375 human melanoma cells and zebrafish embryos. One of the most potent compounds in this series, 19 t, significantly reduced embryonic pigmentation at a concentration of 50 µM, but showed 100% mortality in an acute toxicity test [124].

## 7. Conclusions

Tyrosinase plays a key role in disorders related to depigmentation changes in humans. Thus, TYR inhibitors may be the best option for treatment. Much research has been advancing in the discovery of new inhibitors. A variety of plants and fungi are important producers of bioactive metabolites inhibiting tyrosinase. *Trichoderma* is the most studied genus in terms of tyrosinase inhibition since metabolites of its species are derived from isoflavones and pyrones, along with terpenes, steroids and alkaloids, which can reversibly or irreversibly inactivate the enzyme. In recent years, research has guided important advances in the development of technologies and in the screening of bioactive compounds. Moreover, in vivo tests have intensified the use of the experimental zebrafish model based on phenotypes in which melanin pigments can be observed on the zebrafish surface, allowing the simple observation of the pigmentation process without complicated experimental procedures. For this reason, the zebrafish is gaining increasing viability as an in vivo model to evaluate the depigmenting activity of melanogenic regulatory compounds.

## Figures and Tables

**Figure 1 molecules-28-01053-f001:**
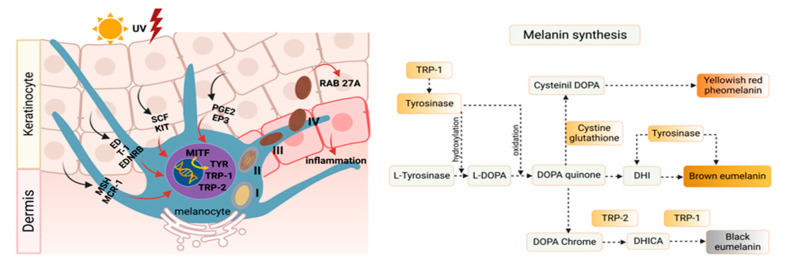
Synthetic pathway of melanin. Melanin synthesis begins with the catalysis of substrates L-phenylalanine and L-tyrosine to produce L-DOPA via phenylalanine hydroxylase (PAH), tyrosinase and, partially, tyrosinase hydroxylase 1 (TH-1). Pathways are then divided into eumelanogenesis or pheomelanogenesis. The other melanogenic enzymes are TRP-2 (DCT) and TRP-1 for eumelanogenesis.

**Figure 2 molecules-28-01053-f002:**
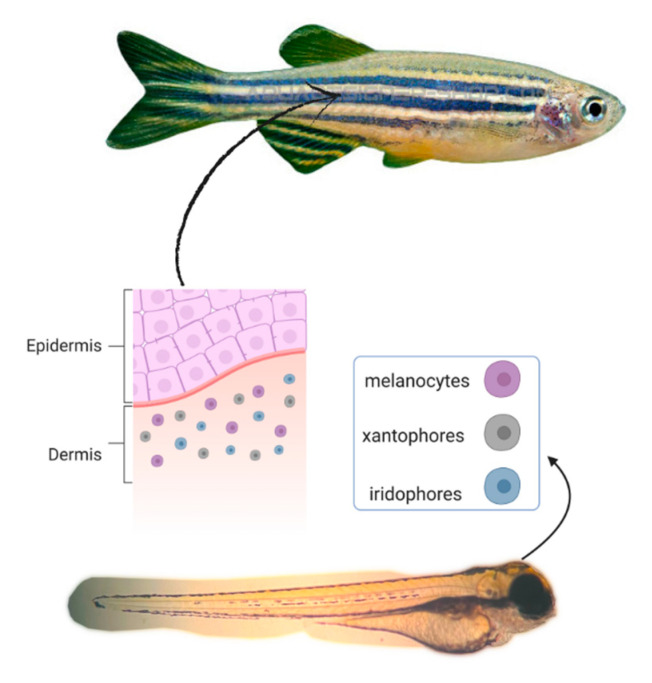
Zebrafish pigment cells include xanthophores (xp), iridophores (ip) and melanophores (mp). The main external barrier is the epidermis, which consists of two layers of cells connected by tight junctions. Certain substances can pass through the epidermis mesenchymal space by diffusion or by active transport (adapted from Irene et al. [67]).

**Figure 3 molecules-28-01053-f003:**
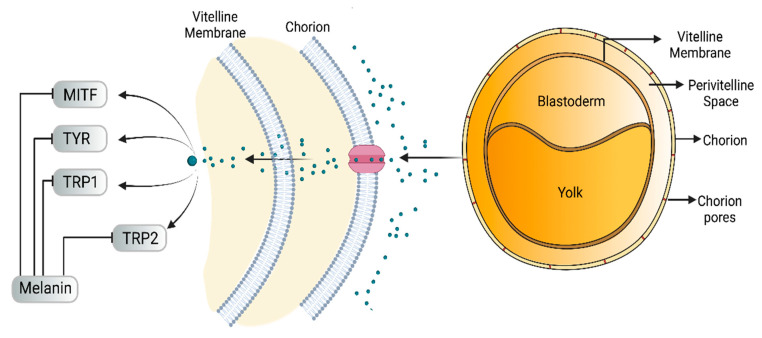
Schematic diagram showing the possible mechanism of action of depigmenting agents on zebrafish embryo. The embryonic chorion is composed of a nanoporous outer membrane 500–700 µM in diameter. The chorion is composed of a three-layer structure (extraembryonic mesoderm) with four main polypeptides. Small or hydrophobic molecules can diffuse across the lipid bilayer (adapted from Bonsignorio et al. [70]; Jon et al. [71]).

**Table 1 molecules-28-01053-t001:** Secondary metabolites found in fungi of the genus *Trichoderma* with anti-melanogenic effect. Structures of tyrosinase inhibitors from *Trichoderma* spp.

Fungus	Molecules and Their Derivatives	Reference
*Trichoderma viride*	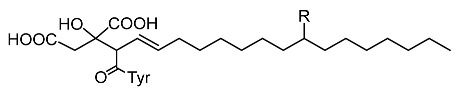 Viridiofungins and derivatives (R = -OH; -H; -C=O)	Reino et al. [53]
*Trichoderma* spp.	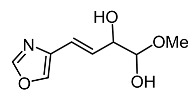 Melanoxazal	Takahashi et al. [54]
*Trichoderma harzianum*	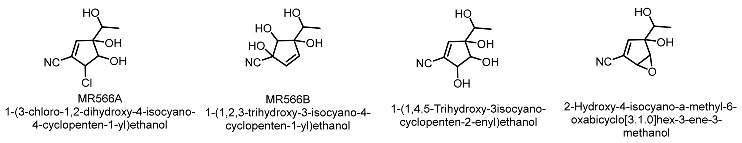	Lee et al. [55]

**Table 2 molecules-28-01053-t002:** Compounds and metabolites derived from plant species used as melanogenesis inhibitors in zebrafish embryos relative to concentration and toxicity.

Entry	Bioactive Compound/Structure	Mechanism	Toxicity/Concentration	Reference
1	Fisetin	Blocks tyrosinase-induced tyrosine oxidation	Did not show (25 µM, 50 µM, 75 µM, and 100 µM)	Ilandarage et al. [61]
2	KDZ-001	TYR active site	Did not show (10 µM)	Kyu-Seok et al. [60]
3	1-phenyl-2-thiourea	Unknown	Did not show	Ilandarage et al. [52]
4	2-mercaptobenzothiazole	Unknown	Did not show	Ilandarage et al. [61] 2020; Tae-Young et al. [66]
5	Haginin	Unknown	Did not show	Tae-Young et al. [66]
6	YT16i	Unknown	Showed toxicity (1 mM)	Tae-Young et al. [66]
7	triclocarban (3,4,4′-trichlorocarbanilide)	Unknown	Showed toxicity (50 µg/L).	Giulia et al. [62]
8	Adenosine	Inhibits melanogenesis by down-regulating tyrosinase	Did not show (400 µM)	Mi Yoon et al. [84]
9	*Ecklonia cava* seaweed extract	Unknown	Slight toxicity (400 µM)	Kang et al. [85]
10	*Sargassum siliquastrum* seaweed extract	Unknown	Did not show (400 µM)	Kang et al. [85]
11	*Ganoderma formosanum* mycelium extract	Blocks tyrosinase-induced tyrosine oxidation	Did not show (400 ppm)	Kai et al. [86]

**Table 3 molecules-28-01053-t003:** Compounds and metabolites derived from plant species used as melanogenesis inhibitors in zebrafish embryos.

Entry	Name	Chemical Structure	Reference
1.	Mearsetin	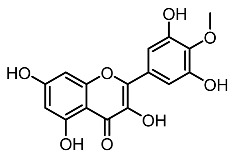	Huang et al. [22]
2.	Myricetin	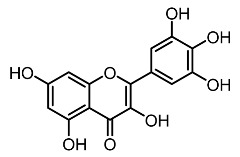	Huang et al. [22]
3.	Arbutin	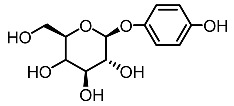	Ilandarage et al. [61]
4.	Niacinamide	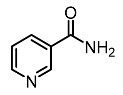	Hako-Zaki et al. [81]
5.	Sesamol	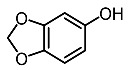	Baek et al. [93]
6.	Gallic acid	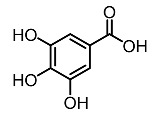	Kumar et al. [94]
7.	Ascorbic acid	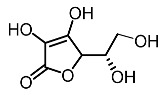	Kumar et al. [94]
8.	Bis(4-hydroxybenzyl)sulfide	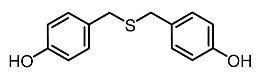	Wang et al. [95]
9.	Coumaric acid	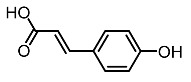	Kim et al. [96]
10.	β-Lapachone	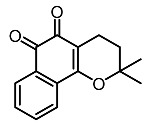	Kim et al. [97]
11.	Tretinoin	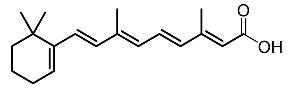	Huang et al. [98]
12.	2-Morpholinobutyl-4-thiophenol	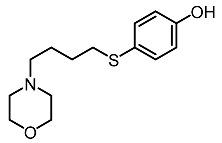	Huang et al. [98]
13.	Biochanin A	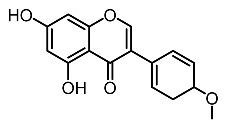	Lin et al. [99]
14.	Subamolide A	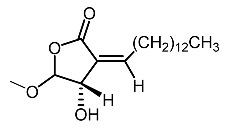	Hiu et al. [100]; Wang et al. [101]
15.	Linderanolide B	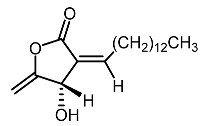	Hiu et al. [100]; Wang et al. [101]
16.	5-Iodotubersidin	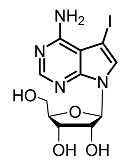	Kim et al. [102]
17.	Glyceollin I	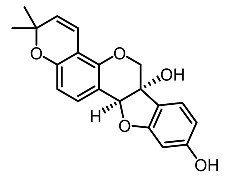	Shin et al. [103]
18.	Arctigenin	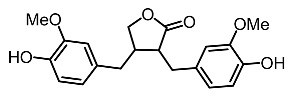	Park et al. [104]
19.	Gomisin N	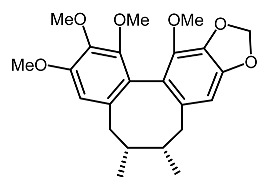	Chae et al. [105]
20.	Haginin A	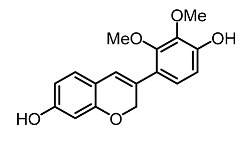	Kim et al. [106]
21.	Glabridin	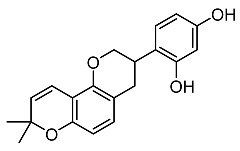	Chen et al. [107]
22.	Floralginsenoside A	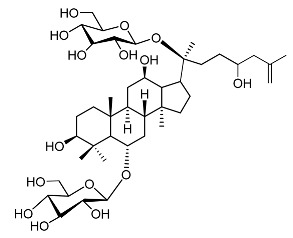	Lee et al. [108]
23.	Ginsenoside Rb2	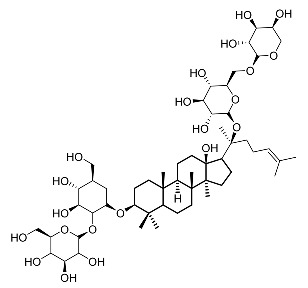	Lee et al. [109].
24.	Octaphlorethol A	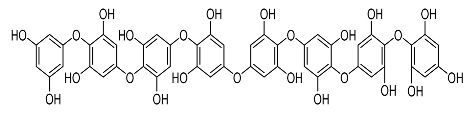	Kin et al. [110]
25.	6-O-isobutyrylbritannilactone	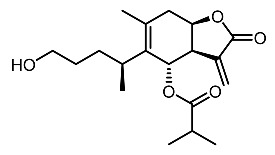	Dae et al. [111]
26.	2,3,7,8-tetrachlorodibenzo-p-dioxin	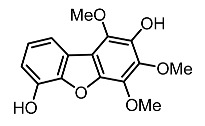	Henry et al. [112]

**Table 4 molecules-28-01053-t004:** Synthetic compounds used as melanogenesis inhibitors in zebrafish embryos.

Entry	Name	Chemical Structure	Reference
9	Sodium erythorbate	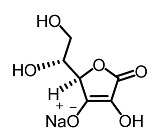	Chen et al. [34]
12	Omeprazole	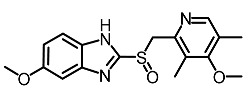	Baek et al. [89]
7	2-Methylphenyl-E-(3-hydroxy- 5-methoxy)-styryl ether	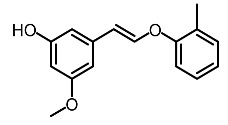	Huang et al. [98]
2	MEK-1	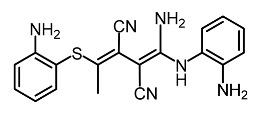	Huang et al. [98]
8	4-phenyl hydroxycoumarins	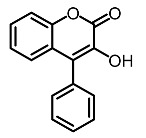	Veselinovi'c et al. [115]
4	Kojic acid palmitate	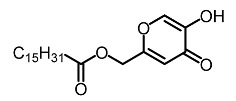	Lajis et al. [116]
1	Suloctidil	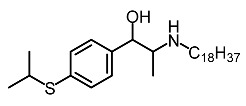	Li et al. [117]
3	Compound 6	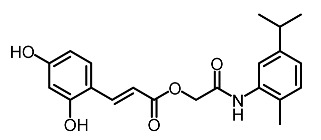	Abbas et al. [118]
5	Fluoxetina	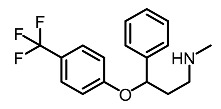	Shang et al. [119]
6	Tretinoína	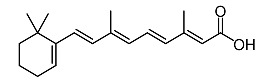	Shang et al. [119]
11	(*E*)-1-(4-(3-chloro-4-fluorophenyl)piperazin-1-yl)-3-(3-nitrophenyl)prop-2-en-1-one	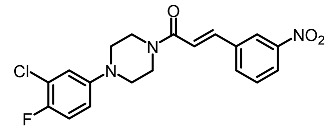	Shang et al. [119]
10	Phenylthiourea	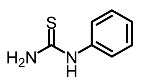	Kim et al. [96]; Hsu et al. [86]; Thach et al. [120]

## Data Availability

Not applicable.

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
