# Peer review of "Anti-Melanogenic Potential of Natural and Synthetic Substances: Application in Zebrafish Model"

_molecules, 2023, doi:10.3390/molecules28031053_

Round 1

Reviewer 1 Report

Overview and general recommendation

            Melanoma is the 17th most common cancer worldwide. In Brazil, they represent 25% of all malignant tumors diagnosed. That is, understanding its mechanism and discovering molecules to inhibit or cure it is of great importance.

However, I missed a greater explanation of how the review was carried out (e.g. database, data collection period, and the type of review that was carried out).

Major comments

1. This work is not an article. It must be submitted as a review, as explained in the instructions for authors (Instructions for Authors).

2. The abstract must bring the conclusion of the authors of the review carried out.

3. Lines 108 – 115. Reference this information.

Author Response

Hanna Xiong

Section Managing Editor

Molecules

We are very pleased to know that our submitted manuscript entitled “Anti-melanogenesis potential of natural and synthetic substances: application in zebrafish model” s was appreciating the time and effort that you and the reviewers dedicated to providing feedback on our manuscript and are grateful for the insightful comments on and valuable improvements to our paper. It incorporated all of the suggestions made by the reviewers. Those changes are highlighted within the manuscript. Please see below, in yellow, for a point-by-point response to the reviewers’ comments and concerns.

R= All manuscript was revised by native in English.

  • Regarding the Request:

Line 19 “in vivo” italic
Line 20 “in vivo” italic
Line 52 “in vivo” italic

R= Authors’ Response: It was carried out as changes as suggested by the reviewer.

  • Regarding the Request:

Line 56 remove the sentence “Initially, tyrosinase is synthesized on the surface of the rough endoplasmic reticulum, 56 considered a key and limiting enzyme for the in vivo melanin synthesis, playing an 57 essential role, with several biological functions, including the pigmentation of the dermis 58 of mammals, present in the primary immune response, and is a trigger of wound healing 59 in plants and fungi. [9]”. It is already present from line 50 to 55.

R=  Authors’ Response: The Paragraph was deleted, as suggested by the reviewer.

  • Regarding the Request:

Line 61 change “In plants, the effect 61 of tyrosinase is observed in the degradation process” to “ In plants, it is observed in degradation processess”

Authors’ Response: The corrections were made for genre writing, as requested by Reviewer.

  • Regarding the Request:

Line 66 the menaing of this sentence in confusing, please change it “In this sense, the discovery of new molecules with therapeutic effect, called New 66 Chemical Entitya (NEQ), from natural products and fungal extracts with anti-67 melanogenesis action, capable of minimizing side effects from the treatment of 68 pigmentation disorders, currently available, thus representing a potent low-cost and 69 effective alternative. [16-19]”

R = Authors’ Response: The wording of the text has been revised and changed for a better understanding of the information..

  • Regarding the Request:

Line 78 “in vivo” and “in vitro” italic

Line 80 “in vivo” italic

R = Authors’ Response: Changes were made as suggested by the reviewer.

  • Regarding the Request:

Line 82 change the sentence “In this sense, emerging models have been used, 82 such as zebrafish (Danio rerio), unlike invertebrate models, zebrafish show high 83 physiological and genetic homology to humans, are easy to manipulate genetically, and 84 display robust behavioral phenotypes.” To “In this sense, emerging models have been used, such as zebrafish (Danio rerio). Unlike invertebrate models, zebrafish show high physiological and genetic homology to humans, are easy to manipulate genetically, and display robust behavioral phenotypes.

R = Text changed conformed suggested.

  • Regarding the Request:

Line 85 change the sentence “With a fully characterized genome with 85 functional domains of many keyproteins nearly identical to their human homologues” to “They are characterized by a fully characterized genome with functional domains of many key proteins nearly identical to their human homologues.

R = Text changed conformed suggested.

  • Regarding the Request:

Line 212 “in vivo” italic

Line 257 “in vivo” italic

Line 301 “in vivo” italic

R = In the lines cited by the reviewer, the word “in vivo” does not appear, however, in lines 304, 383 and 378 the “in vivo” present in the text were revised to italics.

R = All references have again undergone extensive revision, in order to meet the standards of the Journal and the indications of the Reviewer.

Finally, we thank you for the recommendations that were attended to and we are at your disposal for any corrections or doubts about any other points of the work submitted to the Journal.

 Kind regards,

Prof. Dr. Irlon M. Ferreira

Universidade Federal do Amapá

Grupo de Biocatálise e Biotransformação em Química Orgânica
Rod. Juscelino Kubitschek, Km 02, Jardim Marco Zero, Macapá-AP, Brasil, CEP.: 68902-280
Currículo Lattes http://lattes.cnpq.br/9897023410899133

Reviewer 2 Report

No major comments. This is a timely and important review of natural and synthetic molecules with anti-melanogenic properties. The manuscript is well written.

Author Response

Hanna Xiong

Section Managing Editor

Molecules

We are very pleased to know that our submitted manuscript entitled “Anti-melanogenesis potential of natural and synthetic substances: application in zebrafish model” s was appreciating the time and effort that you and the reviewers dedicated to providing feedback on our manuscript and are grateful for the insightful comments on and valuable improvements to our paper. It incorporated all of the suggestions made by the reviewers. Those changes are highlighted within the manuscript. Please see below, in yellow, for a point-by-point response to the reviewers’ comments and concerns.

R= All manuscript was revised by native in English.

  • Regarding the Request:

Line 19 “in vivo” italic
Line 20 “in vivo” italic
Line 52 “in vivo” italic

R= Authors’ Response: It was carried out as changes as suggested by the reviewer.

  • Regarding the Request:

Line 56 remove the sentence “Initially, tyrosinase is synthesized on the surface of the rough endoplasmic reticulum, 56 considered a key and limiting enzyme for the in vivo melanin synthesis, playing an 57 essential role, with several biological functions, including the pigmentation of the dermis 58 of mammals, present in the primary immune response, and is a trigger of wound healing 59 in plants and fungi. [9]”. It is already present from line 50 to 55.

R=  Authors’ Response: The Paragraph was deleted, as suggested by the reviewer.

  • Regarding the Request:

Line 61 change “In plants, the effect 61 of tyrosinase is observed in the degradation process” to “ In plants, it is observed in degradation processess”

Authors’ Response: The corrections were made for genre writing, as requested by Reviewer.

  • Regarding the Request:

Line 66 the menaing of this sentence in confusing, please change it “In this sense, the discovery of new molecules with therapeutic effect, called New 66 Chemical Entitya (NEQ), from natural products and fungal extracts with anti-67 melanogenesis action, capable of minimizing side effects from the treatment of 68 pigmentation disorders, currently available, thus representing a potent low-cost and 69 effective alternative. [16-19]”

R = Authors’ Response: The wording of the text has been revised and changed for a better understanding of the information..

  • Regarding the Request:

Line 78 “in vivo” and “in vitro” italic

Line 80 “in vivo” italic

R = Authors’ Response: Changes were made as suggested by the reviewer.

  • Regarding the Request:

Line 82 change the sentence “In this sense, emerging models have been used, 82 such as zebrafish (Danio rerio), unlike invertebrate models, zebrafish show high 83 physiological and genetic homology to humans, are easy to manipulate genetically, and 84 display robust behavioral phenotypes.” To “In this sense, emerging models have been used, such as zebrafish (Danio rerio). Unlike invertebrate models, zebrafish show high physiological and genetic homology to humans, are easy to manipulate genetically, and display robust behavioral phenotypes.

R = Text changed conformed suggested.

  • Regarding the Request:

Line 85 change the sentence “With a fully characterized genome with 85 functional domains of many keyproteins nearly identical to their human homologues” to “They are characterized by a fully characterized genome with functional domains of many key proteins nearly identical to their human homologues.

R = Text changed conformed suggested.

  • Regarding the Request:

Line 212 “in vivo” italic

Line 257 “in vivo” italic

Linha 301 "in vivo" itálico

R = Nas linhas citadas pelo revisor, a palavra "in vivo" não aparece, porém, nas linhas 304, 383 e 378 o "in vivo" presente no texto foi revisado para itálico.

R = Todas as referências passaram novamente por uma extensa revisão, a fim de atender aos padrões da Revista e às indicações do Revisor.

Por fim, agradecemos as recomendações atendidas e estamos à disposição para quaisquer correções ou dúvidas sobre quaisquer outros pontos do trabalho submetido à Revista.

Saudações

Prof. Dr. Irlon M. Ferreira

Universidade Federal do Amapá

Grupo de Biocatálise e Biotransformação em Química Orgânica
Rod. Juscelino Kubitschek, Km 02, Jardim Marco Zero, Macapá-AP, Brasil, CEP.: 68902-280
Currículo Lattes http://lattes.cnpq.br/9897023410899133

Author Response

(The authors gave the same response as above.)

Round 2

Reviewer 1 Report

All reviewers' comments and suggestions were addressed.

Author Response

Hanna Xiong

Section Managing Editor

Molecules

We are very pleased to know that our submitted manuscript entitled “Anti-melanogenesis potential of natural and synthetic substances: application in zebrafish model” s was appreciating the time and effort that you and the reviewers dedicated to providing feedback on our manuscript and are grateful for the insightful comments on and valuable improvements to our paper. It incorporated all of the suggestions made by the reviewers. Those changes are highlighted within the manuscript. Please see below, in yellow, for a point-by-point response to the reviewers’ comments and concerns.

R= All manuscript was revised by native in English.

  • Regarding the Request:

Line 19 “in vivo” italic
Line 20 “in vivo” italic
Line 52 “in vivo” italic

R= Authors’ Response: It was carried out as changes as suggested by the reviewer.

  • Regarding the Request:

Line 56 remove the sentence “Initially, tyrosinase is synthesized on the surface of the rough endoplasmic reticulum, 56 considered a key and limiting enzyme for the in vivo melanin synthesis, playing an 57 essential role, with several biological functions, including the pigmentation of the dermis 58 of mammals, present in the primary immune response, and is a trigger of wound healing 59 in plants and fungi. [9]”. It is already present from line 50 to 55.

R=  Authors’ Response: The Paragraph was deleted, as suggested by the reviewer.

  • Regarding the Request:

Line 61 change “In plants, the effect 61 of tyrosinase is observed in the degradation process” to “ In plants, it is observed in degradation processess”

Authors’ Response: The corrections were made for genre writing, as requested by Reviewer.

  • Regarding the Request:

Line 66 the menaing of this sentence in confusing, please change it “In this sense, the discovery of new molecules with therapeutic effect, called New 66 Chemical Entitya (NEQ), from natural products and fungal extracts with anti-67 melanogenesis action, capable of minimizing side effects from the treatment of 68 pigmentation disorders, currently available, thus representing a potent low-cost and 69 effective alternative. [16-19]”

R = Authors’ Response: The wording of the text has been revised and changed for a better understanding of the information..

  • Regarding the Request:

Line 78 “in vivo” and “in vitro” italic

Line 80 “in vivo” italic

R = Authors’ Response: Changes were made as suggested by the reviewer.

  • Regarding the Request:

Line 82 change the sentence “In this sense, emerging models have been used, 82 such as zebrafish (Danio rerio), unlike invertebrate models, zebrafish show high 83 physiological and genetic homology to humans, are easy to manipulate genetically, and 84 display robust behavioral phenotypes.” To “In this sense, emerging models have been used, such as zebrafish (Danio rerio). Unlike invertebrate models, zebrafish show high physiological and genetic homology to humans, are easy to manipulate genetically, and display robust behavioral phenotypes.

R = Text changed conformed suggested.

  • Regarding the Request:

Line 85 change the sentence “With a fully characterized genome with 85 functional domains of many keyproteins nearly identical to their human homologues” to “They are characterized by a fully characterized genome with functional domains of many key proteins nearly identical to their human homologues.

R = Text changed conformed suggested.

  • Regarding the Request:

Line 212 “in vivo” italic

Line 257 “in vivo” italic

Line 301 “in vivo” italic

R = In the lines cited by the reviewer, the word “in vivo” does not appear, however, in lines 304, 383 and 378 the “in vivo” present in the text were revised to italics.

  1. Table 3, item 20, please double check the drawing for the structure.

R = Table changed conformed suggested.

  1. Table 3, item 23, please double check the drawing for the structure. It cannot be displayed properly.

R = Table changed conformed suggested.

  1. Please double check again for the chemical structures to make sure all are presented correctly.

R = Chemical structure checked conformed suggested.

  1. Page 16, lines 350-355, the labels, 19p and 19t, are not found for the compounds.

R = Text changed conformed suggested.

  1. "Reference" and "Refs" should be kept consistently used in Table 1-3.  

R = Text changed conformed suggested.

  1. Is this manuscript submitted as a Review Article?

R = Yes, as a Review.

  1. Please re-structure the Conclusion Section into one paragraph.

R = Text revised conformed suggested.

R = All references have again undergone extensive revision, in order to meet the standards of the Journal and the indications of the Reviewer.

Finally, we thank you for the recommendations that were attended to and we are at your disposal for any corrections or doubts about any other points of the work submitted to the Journal.

 Kind regards,

Prof. Dr. Irlon M. Ferreira

Universidade Federal do Amapá

Grupo de Biocatálise e Biotransformação em Química Orgânica
Rod. Juscelino Kubitschek, Km 02, Jardim Marco Zero, Macapá-AP, Brasil, CEP.: 68902-280
Currículo Lattes http://lattes.cnpq.br/9897023410899133
